# Profiling the Physiological Roles in Fish Primary Cell Culture

**DOI:** 10.3390/biology12121454

**Published:** 2023-11-21

**Authors:** Lingjie He, Cheng Zhao, Qi Xiao, Ju Zhao, Haifeng Liu, Jun Jiang, Quanquan Cao

**Affiliations:** 1College of Animal Science and Technology, Sichuan Agricultural University, Chengdu 611130, China; 2022202078@stu.sicau.edu.cn (L.H.); 2020202064@stu.sicau.edu.cn (Q.X.); 2020302174@stu.sicau.edu.cn (J.Z.); liuhf@sicau.edu.cn (H.L.); 2College of Marine Science and Engineering, Jiangsu Province Engineering Research Center for Aquatic Animals Breeding and Green Efficient Aquacultural Technology, Nanjing Normal University, Nanjing 210023, China; 90844@njnu.edu.cn

**Keywords:** primary fish cell culture, fish cell line, fisheries science, aquaculture

## Abstract

**Simple Summary:**

The field of cell culture technology involves the cultivation of cells in a controlled environment, mimicking the natural conditions found in living organisms. This process is performed on a large scale, utilizing a specially designed culture medium. The goal is to create either undifferentiated individual cells or multicellular aggregates with minimal differentiation. The present paper offers a comprehensive overview of the advancements and applications of primary cell culture techniques in fish. It places a particular focus on revealing the physiological roles played by fish cells during their in vitro cultivation. Maintaining the functional characteristics of fish cells is of paramount importance, as it is essential for gaining a deeper understanding of the underlying mechanisms governing various physiological processes. To facilitate the use of fish cells in research and practical applications, it is crucial to standardize the separation techniques for these cells and optimize the conditions for their culture. This not only contributes to our knowledge of fish health, disease development, and drug discovery but also drives progress in the aquaculture industry.

**Abstract:**

Fish primary cell culture has emerged as a valuable tool for investigating the physiological roles and responses of various cell types found in fish species. This review aims to provide an overview of the advancements and applications of fish primary cell culture techniques, focusing on the profiling of physiological roles exhibited by fish cells in vitro. Fish primary cell culture involves the isolation and cultivation of cells directly derived from fish tissues, maintaining their functional characteristics and enabling researchers to study their behavior and responses under controlled conditions. Over the years, significant progress has been made in optimizing the culture conditions, establishing standardized protocols, and improving the characterization techniques for fish primary cell cultures. The review highlights the diverse cell types that have been successfully cultured from different fish species, including gonad cells, pituitary cells, muscle cells, hepatocytes, kidney and immune cells, adipocyte cells and myeloid cells, brain cells, primary fin cells, gill cells, and other cells. Each cell type exhibits distinct physiological functions, contributing to vital processes such as metabolism, tissue regeneration, immune response, and toxin metabolism. Furthermore, this paper explores the pivotal role of fish primary cell culture in elucidating the mechanisms underlying various physiological processes. Researchers have utilized fish primary cell cultures to study the effects of environmental factors, toxins, pathogens, and pharmaceutical compounds on cellular functions, providing valuable insights into fish health, disease pathogenesis, and drug development. The paper also discusses the application of fish primary cell cultures in aquaculture research, particularly in investigating fish growth, nutrition, reproduction, and stress responses. By mimicking the in vivo conditions in vitro, primary cell culture has proven instrumental in identifying key factors influencing fish health and performance, thereby contributing to the development of sustainable aquaculture practices.

## 1. Introduction

The field of cell culture technology has its roots in the early 20th century and, since then, has boasted a rich history that spans over a century. Research into primary cell culture techniques for fish began in the middle of the last century. Notably, the first permanent cell line of fish origin was pioneered by Wolf and Quimby in 1962, using gonadal cells from rainbow trout (*Oncorhynchus mykiss*) [1]. Throughout the mid-20th century, there was a significant increase in the availability of tissues suitable for primary cell culture. This development paved the way for successful applications of various fish tissues, including liver, spleen, gonads, fin strips, skin, and fish cells in the fields of immune system research, vaccine research, and biomedical research. For example, primary culture of zebrafish embryonic stem cells provided a means to examine retinal pigment differentiation and function prior to optic organogenesis [2]. Fish cell cultures have multiple scientific objectives, such as study of virus diagnosis skills, the study of the thermoregulation mechanism, endocrinological investigation, and viral disease diagnosis [3,4]. They can also be used as a suitable model for evaluating the oxidation reaction, chemistry toxicology tests, and mechanism research [5]. In environmental science, fish cells are in use for detecting the impacts of environmental pollutants and toxicology of aquatic ecosystems [4,6].

Primary cell culture has many scientific superiorities and can be kept for weeks, so it can be used for vitro detecting systems [7]. Primary cells have high viability and some essential cellular functions can be maintained in vitro such as cell proliferation and differentiation [8]. For example, detached testis cells were gathered in one day and isolated into colonies, which adhered to Petri medium or carried out suspended growth [9]. Under selected experimental conditions, researchers can culture the cells of interest in vitro and study them at a cellular level. They can survive and retain specific characteristics of the cell type for at least a few days in many cases [9]. Alternatively, primary cells can retain more original functions and abilities compared with cell lines with limited ability [10]. Primary cells and cell lines both have advantages and disadvantages. The growth rate of primary cells is notably slow, and conventionally, the initial generation of primary cell culture and its subsequent passages up to the 10th generation collectively form the primary culture. Beyond approximately 10 generations, primary cultured cells present a formidable challenge for propagation. Cellular growth tends to stagnate, and most cells undergo senescence and apoptosis. Nevertheless, a small fraction of cells somehow endure this crisis, persisting in propagation. Some of these cells may even undergo genetic alterations and acquire malignant characteristics, allowing for indefinite passageability, which is referred to as a cell line. Primary cells, isolated directly from animal tissues through enzymatic or mechanical methods, are often considered more biologically relevant than cell lines. This is because their biological responses may closely mimic in vivo conditions, bringing researchers closer to the “truth” [11]. Early cultures of primary cells can offer a more accurate model of in vivo tissues [12]. However, the production of short-term primary cultures faces challenges related to the reproducibility of initiation and the homogeneity of cultures, limiting their applications [13]. The advantages of easy culture, diversity, and substantial yield have positioned cell lines as the preferred choice for cellular-level research. Nevertheless, it is important to note that continuous culturing of cell lines can lead to mutations over time, potentially altering both their genotype and phenotype, which in turn may impact experimental outcomes. Primary cells do not encounter these problems. For instance, cells obtained from EK293 transfected with adenovirus exhibited characteristics closer to immature neurons [14]. The MDA-435 cell line, long utilized as a representative model for breast cancer, has recently raised questions about its true identity, with accumulating evidence suggesting its potential classification as a melanoma cell [15]. In summary, for experiments that cannot be conducted within living organisms, primary cells can maintain a high level of biological characteristics and thus offer a partial solution to this issue (Table 1). Vitro cell culture is a fascinating experiment method as the cells can maintain their interaction, polarity, and topology [7]. The main benefit is that the primary fish cells would be reassembled into a single-layer structure, which can represent the most primitive state of the living fish instead of complex tissues. Moreover, in vitro cell culture has a very low cost as cell care does not require large farming facilities and large amounts of water compared to live fish [5]. 

The principles and techniques of fish cell culture draw heavily from mammalian cell culture practices. However, when compared to mammalian cell culture, fish cell culture presents some distinct characteristics. Notably, fish cell culture boasts several advantages including its ability to adapt to a wide range of temperatures, higher tolerance to hypoxia, and the ease of maintaining cell cultures for extended durations [16]. In particular, in physiology, fish cell culture can provide a perfect instrument for studying the response of cells to different culture temperatures because fish primary cell culture can simulate living conditions in vivo [17]. In immunology, cell cultures are indispensable in allowing viral transmission and they make a significant contribution to the development of vaccines to prevent multiple fish diseases [18]. In ecotoxicology, the utility of fish cell culture also has unrivaled benefits, such as quick revolution, convenience, ease of identification, effectiveness, and high specificity compared with vivo organism testing [3]. Technically and scientifically, vitro cell culture techniques can screen many specimens for a reasonable cost, especially for environment toxicant assays [4]. Furthermore, primary fish cell cultures can reduce the number of animals killed, because it is moral to kill fewer fish [3]. Extrinsic cytotoxicity testing has the advantage of being able to quickly detect the potential virulent concentration of specific substances, for instance, chemical or nanomaterial. This is an excellent toxicity grading and grouping tool that can be used as a pre-screening intelligent detection strategy to help regulators make decisions and reduce in vivo testing [19]. It is also an available instrument to measure the poisoning of various marine pollutants [4]. Recently, fish have emerged as a suitable model and a promising alternative to the classical mammalian systems to study vertebrate development.

The growing economic significance of aquaculture has driven increased research into the healthy development and pathological occurrences of fish, with particular attention being paid to the prevention and control of freshwater and seawater diseases, as well as environmental issues. The advancement of cell culture technology has played a pivotal role in enabling research in genomics, the study of virus–host interaction studies, bacterial identification, heavy metal toxicity analysis, and stem cell function research. To our knowledge, no review has summarized the information on primary cell cultures from different fish tissues. Thus, we report on the progress of primary cell cultures derived from the gonad, pituitary, muscle, liver, kidney, adipocyte, myeloid, brain, fin, and gill and the historical background, practical applications, and cultivation techniques employed in fish cell culture in this review (Figure 1). It is expected to provide some valuable references for the development of fish primary cell culture and the establishment of cell lines.

## 2. Primary Cells of Each Tissue

### 2.1. Fish Gonad Cells

The major cell types of fish sexual gland cells are gonad cells and surrounding somatic cells. The main function of the surrounding somatic cells is to produce developmental gonad cells and to synthesize relevant steroids to adjust the development and maturation of oocytes and spermatocytes [13]. The interaction between gonad cells and surrounding somatic cells may play a role during the development of functional sex and developmental gonad cells of fish [20]. In teleosts, ovarian stem cells are derived from the epithelium of the ovary, and hormones ejected by the pituitary can influence the follicles’ growth, development, and maturity [21]. In the male testis, local ligands, deriving from multifarious testicular germ cells, especially for growth factors, can regulate and modify the function of testicular support cells and mesenchymal cells through spermatogenesis. In the female ovary, ovary membrane cells are important for regulating and producing androgens and aromatase from vitro granulosa cells [10].

According to morphological observation, follicular cells and testicular cells are composed of different types of cells from gonadal tissues. Primary gonadal cells (testicular cells) from large yellow crocea (*Larimichthys crocea*) were cultured for study [22]. This system was used to be closer to the body than cell lines, because different paracrine factors in various cells can be presented in the endocrine system [10]. An in vitro test system using primary testis cells of the medaka (*Oryzias latipes*) was established that provides quantitative data on cell proliferation and cell viability [9]. Primary cell cultures (ovarian follicular and testicular cells) of the marine medaka (*Oryzias melastigma*) were used to screen the environmentally relevant levels of endocrine-disrupting chemicals [10]. Primary sperm cultures can be regarded as an adaptive detection system to analyze endocrine disruptors and other substances suspected of interfering with fish sperm formation [7]. Flow cytometry of sperm cells can be easily quantified to explore the various cell indexes, including cellular size and cellular ploidy. Therefore, the technique is widely used in the research of spermatogenesis covering the kinetics of cellular proliferation on the basis of the continuous decline of fluorescence density in each cleavage cell [9]. Primary cell culture in the ovary of medaka was used to show that nanosilver can have effects on hormone production and cell apoptosis in teleosts. The in vitro model was used to evaluate endocrine disruption and toxicity [23].

### 2.2. Fish Pituitary Cells

Pituitary cells are mainly related to reproduction, growth, and stress response [10]. The pituitary displays different families of structurally and functionally related adenohypophyseal hormones: single-chain polypeptide and glycoprotein hormones (growth hormone (GH); prolactin (PRL); somatolactin (SL); gonadotropins (GTH); and thyroid-stimulating hormone (TSH)) and proopiomelanocortin-derived hormones (adrenocorticotropic (ACTH) and melanophore-stimulating hormone (MSH)) [24]. Atlantic cod’s primary pituitary cell culture was used to study the underlying immediate impacts of cortisol. It provided evidence that cortisol stimulates cellular activity and affects reproductive expression [25]. Cortisol stimulates the release of growth hormone through the nitric oxide pathways and the degradation pathways of guanine nucleotides (GMP routes) independent pathways [26]. The GMP routes do not participate in the suppression of growth hormone release in a grouper pituitary cell culture platform. Irisin directly inhibits the gene expression and secretion of growth hormone when using primary tilapia pituitary cells. And atropin could stimulate the expression of genes related to growth hormone via direct action at the pituitary level [27,28]. 

Cod’s primary pituitary cells were used in electro-physiological experiments and culturing conditions were optimized: cells that were incubated in M199 medium with osmolality of 320 mOsm were close to the natural plasma osmolality of cod; 26.2 mM HCO_3_^−1^ was used in M199 medium and the CO_2_ concentration of the incubating environment was set as 0.5% which led to the physiological pH of 7.85. Serum was necessary for pituitary cell culture and bovine serum albumin should be supplemented. This serum substitute should contain components that keep the structure and plasma membrane intact because they can influence the cell physiology [29]. The optimized culture conditions for medaka (*Oryzias latipes*) pituitary primary cells were established and investigated [12].

### 2.3. Fish Muscle Cells

Fish muscles are of particular importance from a practical aquaculture view because the weight of fish muscles determines the yield of aquaculture. So, understanding muscle cells is essential for aquaculture to increase fish production [30]. Trout myocyte culture was explored in 1995, optimized in 1998, and regarded as an available tool to analyze the rule of insulin-like growth factor (IGF) in muscle cells of trout in 2002 [31]. Primary muscle cell culture was incorporated at an osmolarity of 360 mOsm and a temperature of 15–21 °C for trout and sea bream [31]. F10 medium was better for muscle cell culture to control in vitro myoblast proliferation and differentiation [32]. The muscle tissue cells from grass goldfish were primary cultured at various time points after seeding [33]. Primary trout muscle cell culture was performed to explore the transcriptional regulation mechanisms of autophagy-related genes after food starvation [30]. Primary muscle cell culture of gilthead sea bream was developed to characterize insulin growth factor-1 binding during myocyte differentiation, which is involved in the role of insulin in metabolism and growth of muscle cells of black bream [31]. Primary brown trout muscle cells can replicate the differentiation process in skeletal muscle and have been used to study the direct metabolism of hormones and cytokines in the muscle of trout [34]. The extracted muscle cells were characterized to validate the role of myogenesis in trout [32].

### 2.4. Fish Hepatocytes

Primary fish liver cells have been shown to be multifunctional and the primary cell culture of liver is widely used for donor organs. In primary culture, fish liver cells can maintain the natural characteristics of liver, and primary liver cells are used for biotransformation, detoxification, and fat generation [35]. Liver cells are commonly used in the study of ecotoxicology. Hepatocytes have been shown to be of use for toxicological screening of cytotoxicity, endocrine disruption, and bioaccumulation through a variety of tests including suspension, monolayer, and three-dimensional sphere cultures [36]. Primary hepatocytes were extracted from orange grouper to evaluate the toxic effects of nonylphenol on hepatocyte activity and the antioxidant system in the liver. Nonylphenol was found to affect the balance of antioxidant defense and lead to the oxidation imbalance of primary hepatocytes [3]. Primary hepatocytes were isolated successfully from arctic char. The optimization of culture conditions to determine the analytical conditions provides a common tool for screening the potential impact of contaminants and complex samples’ exposure to chemicals [35]. Brown trout hepatocytes in an in vitro model were linked to the physiological and toxicological effects of nuclear receptors in cross-regulation and cross-interference of the peroxidase pathway [37]. Primary hepatocytes can be used as a sensitivity test system to measure the cytotoxicity of nanomaterials [38]. Copper nanoparticles were toxic to primary grouper hepatocytes, but the toxicity of CuSO_4_ was more severe than that of copper nanoparticles (Cu NPs) [39]. It was reported that benzopyrene had the potential to induce oxidative stress in primary grouper liver cells and the oxidative effects of benzopyrene were assessed as a potential toxicology research technique [4]. Fish liver cell lines were also widely used in vitro models to study the adverse effects of pollutants in the marine environment [40]. The use of a 3-D in vitro liver organoid culture system (spheroids) derived from rainbow trout was reported to measure the metabolism of pharmaceuticals using a substrate depletion assay. Hepatocyte isolation and cell culture from Lake Van fish have been successfully achieved [41]. Liver spheroids could be used as a relevant and metabolically competent in vitro model to measure the biotransformation of pharmaceuticals in fish [42].

Primary fish hepatocytes have been considered as a model for functional studies of fish. For example, using a primary culture of fish hepatocytes, the effect of insulin on fat generation and lipid hydrolysis can help to understand the regulation of lipid metabolism in vivo at transcription and enzyme levels [43]. Furthermore, palmitic acid treatment in large yellow croaker hepatocytes was used to mimic the effect of feeds for farmed fish and it was found that hepatic cellular triacylglycerol accumulation was significantly higher in palmitic acid treatment [44]. One method was successfully established in trout liver cells to silence the activity of peroxidase proliferator-activated receptor (gene silencing efficiency was about 70%) and showed morphological characteristics of the phenotype after silencing [45]. 

RNA interference (RNAi) was performed to knock down a STAT family member (stat5bl) using primary liver cell culture and could regulate the JAK/STAT pathway to improve fish immunity [46]. Detection of pH regulation ability in primary sturgeon hepatocytes can be used to assess the suitability of the cellular mechanism of CO_2_ tolerance in sturgeon, and primary sturgeon liver cells can compensate for intracellular acidosis caused by hypercapnia [47]. A study showed that during isolation and culture of primary hepatocytes from yellow catfish (*Pelteobagrus fulvidraco*), Zn attenuated Cu-induced lipotoxicity by reducing lipogenesis and stimulating lipolysis [48], and the hepatocytes were incubated with ZnO NPs (10 μg/mL) to investigate the mechanisms by which ZnO NPs influence the Zn absorption and lipid metabolism [49].

### 2.5. Fish Kidney and Immune Cells

Fish kidneys are typically divided into two parts: the head kidney and the posterior kidney. The primary function of the posterior kidney is to regulate water balance, excrete divalent ions, and manage metabolic processes [50]. Interestingly, this aspect is seldom discussed in the literature. On the other hand, the head kidney primarily plays a crucial role in the production of macrophages in fish. These macrophages, derived from the head kidney, exhibit the capacity to phagocytize, produce radicals, and polarize into either innately activated or alternatively activated macrophages [51]. Consequently, they play a pivotal role in both the innate and acquired immune responses of fish [52]. In an exciting development, a continuous cell line has been established, characterized, and isolated from the head kidney of a large yellow croaker. This cell line holds significant potential as a valuable tool for studying immune-related genes and functions [53]. Furthermore, a study involving seabass head kidney in vitro research has shed light on the immunomodulatory effects of various amino acids. This underscores the potential for developing an immune nutrition strategy [54]. The aquaculture industry has suffered significant economic losses due to bacterial and viral infections. In order to mitigate these losses, cell cultures offer a viable alternative to in vivo experimentation. A study was conducted on kidney primary cell cultures from three Chilean salmonids, namely *Salmo salar*, *Oncorhynchus kisutch*, and *Oncorhynchus mykiss*. The aim of the study was to characterize the response to bacterial and viral stimuli by evaluating various markers associated with both innate and adaptive immune responses [55]. In another study, the time course of stimulation with growth hormone (GH) and growth hormone release factor (GRF) was described in two experimental models: the first model involved the SHK-1 cell line derived from primary cultures of adherent cells from Atlantic salmon (*Salmo salar*) head kidney, which exhibited phagocytic characteristics; and the second model utilized leukocytes isolated from the head kidney of Atlantic salmon. The results indicated differential regulation between these two models, providing a better understanding of the independent action of GH on the immune system [56]. In the head kidney cellular primary cell culture of sea bream and rainbow trout, cortisol had a prominent influence on the expression of major inflammatory cytokines [57]. Furthermore, corticotropin caused a distinct regulation of cytokines in the expression of proinflammatory cytokines in the head kidney cells of rainbow trout and sea bream, which suggested that stress hormones have different regulatory effects on the immune response of teleosts [58]. A protocol of primary wolf fish kidney monocytic cells was used to develop and evaluate the toxicity by exposure to different NSAIDs (Nonsteroidal Anti-inflammatory Drugs) [59].

Phagocytes are the body’s first line of defense against foreign invasion and the bridge between the body’s innate immune system and the adaptive immune system [60]. In this case, macrophages are key immune cells in the early stages of infection as they not only secrete cytokines to regulate the activation and migration of other white blood cells but also occupy an important role in phagocytosis and antigen recognition [61]. Macrophages are some of the most significant effector cells in the natural immune system. The ability to swallow pathogens and coordinate immune responses depends on the presence of different surface receptors, such as scavenger receptors and Toll-like receptors [6]. Primary head kidney macrophages were isolated from red carp, the immune regulatory effects were investigated, and immune systems were simplified and improved [5]. The functional characterization of interleukin and its role in the induction of primary renal macrophages in goldfish was confirmed and the conservatism of interleukin-induced selective activation of macrophages and the importance of polarization of macrophages in the evolution of epigenetic animals were emphasized [62]. Eel primary macrophages were developed and described from head kidney in vitro. The phagocytic activity was measured in the derived cells of head kidney, and the cell culture system offered an important resource for identifying molecular tools and useful models for studying the interactions between specific eel pathogens [6]. Moreover, respiratory burst and phagocytic activity were evaluated by primary culture of renal leukocytes [60]. N6-cyclohexyl played a down-regulating role in the innate immune function of white blood cells in the head kidney, while adenosine did not down-regulate, suggesting the existence of purine receptors in fish immune cells [63]. 

### 2.6. Fish Adipocyte Cells and Myeloid Cells

Superfluous accumulation of fat tissue in cultured fish is a prominent problem in fish aquaculture. A continued intensification in aquaculture can lead to the development of calorie-rich food, which can lead to an increase in subcutaneous and visceral fat. Such obesity can have a negative effect on fish health. Primary culture of fish fat cells has been proven to be an effective method to detect the effects of hormones on fat production and metabolism [64]. The primary adipocytes of red sea bream were cultivated using an induction medium [65].

By comparing the effects of organotin on adipocyte development and lipid metabolism in rainbow trout cell culture, the main use of adipocyte culture was as a valuable tool to estimate the ability of different compounds to interfere with adipocyte differentiation and lipid accumulation [66]. The effects of leptin and ghrelin on the expression of genes related to lipogenesis, lipolysis, and lipid metabolism in rainbow trout fat cells were investigated [67]. Rainbow trout adipocyte cells were applied for coordinating the expression of related lipid-droplet genes in proliferation and differentiation, which occupies a critical role in specific stages of fish fat formation [68].

Adipocyte cells and bone cells were often studied in previous research. The adipocyte and bone-derived cells of sea bream can be differentiated into adipocyte-like cells after adding the differentiation medium. Mesenchymal stem cells are from fatty tissue or vertebrae of sea bream. The expression profiles of genes and transcription factors were studied to relate them to lipid metabolism during lipogenesis [69]. Myeloid cells, including macrophages, neutrophils, granulocytes, thrombocytes, and erythrocytes, which are from a small number of hematopoietic stem cells or hematopoietic progenitor cells (Figure 2) are collectively related to myelopoiesis [70]. Primary cell cultures have been established from sea bream. During culture, mineral deposition in extracellular matrix can promote the addition of osteogenic medium [71]. Previous studies were conducted on the process of bone formation in fish. Insulin-like growth factor (IGF) and insulin were identified as regulators of bone marrow cell proliferation, which has a great significance for better understanding the development of bone growth and bone deformity of fish and improving the quality of aquaculture products in the future [31]. 

### 2.7. Fish Brain Cells

The cell types of brains can be divided into microglia, neurons, and astroglia. Primary brain cell culture was performed for giant grouper. Interleukin-1 and tumor necrosis factor (TNF) were found to cause the death of neurons in brain cells after nervous necrosis virus infection [72]. The effect of kisspeptin decapeptide on the expression of reproduction-related genes was evaluated using the primary hypothalamic culture system, and the results showed that kisspeptin had no effect on gonadotropin-releasing hormone in the brain cells of tongue sole [44]. Primary neuron cultures of Senegalese sole were established and analyzed using a detailed protocol, which proved the effects of two capsid mutations in recombinant strains on virus replication of nerve cells [73]. A continuous cell line from the brain of perch was susceptible to necrosis and the culture system provided a useful tool to study epidemiology, viral pathology, and vaccine development [74]. Cell lines of grouper brain tissue were established and their susceptibility was studied, including to red-spot grouper neuro-necrosis virus, Singapore grouper iris virus, and lymphocytosis virus [75].

### 2.8. Fish Gill Cells

Gills are the main organs for fish to exchange gas and ions with the external environment. Branchial cells have a complex epithelial tissue composed of three main types of cells in direct contact with the environment: pavement cells (also known as respiratory cells), mitochondria-rich cells (also known as chlorine cells), and mucous cells [76]. Branchial ion extrusion is thought to occur between columnar or elliptical ionic cells, known as mitochondria-rich cells (MRCs) and adjacent cells referred to as thin-laminar adherent cells (ACs). Na^+^ enters the apical fossa of MRCs through “leaky” paracellular connections between MRCs and interlaced ACs [77]. The pavement cells mainly cover the chloride ion cells on the branchial filaments and branchial lamella. The mucous cells are in the lamella epithelium and their secretion may be a mechanism to adapt to different water conditions [78]. A study identified the hypotonic responsive genes in gill cells and profiled the gill microbiota communities after a freshwater transfer experiment via transcriptome sequencing and 16S rRNA gene sequencing, suggesting the host–bacterium interaction in the gill facilitates freshwater acclimation [17].

The establishment of the freshwater teleost gill model was based on the reconstruction of squamous epithelium on the permeable membrane of primary culture [79]. In vitro culture of gill epithelium has two unique advantages. The first advantage is the use of freshly isolated gill cells to obtain information on ion transport in filament surfaces and MRCs to explain the osmotic mechanism. The second advantage is that gill filaments can be cultured in a culture medium for almost four days [78]. The Percoll method is the best method for gill cell isolation; Percoll_TM_-gradient-isolated gill chloride (CC) and pavement cells (PVCs) were settled and the regulation of their expression in primary gill cell culture was determined [80,81,82]. The primary culture of gill epithelial cells from rainbow trout (*Oncorhynchus mykiss*) has been successfully investigated [83]. The primary culture of fish gill cells can provide a functional, cell-diversified and model culture platform in vitro. In a study to prolong the viability of primary gill cell cultures in rainbow trout, a method was established to prolong the viability of cultures during prolonged exposure to water, with the utility of this model extending to a variety of longer-term exposure scenarios [82].

Gill cells are mainly studied for osmoregulation. Previous studies have revealed that the cascade of osmotic signals may be related to the regulation of downstream effects [84]. Mitogen-activated protein kinases and myosin light-chain kinases were explored in the early phase of hyperosmotic challenge using primary freshwater teleost branchial cell culture [85]. Short-term hyperosmotic challenges can activate multiple severalsignaling pathways. For example, histone phosphorylation and downstream effectors played a role in freshwater gill cells and hypertonicity-challenge-induced MAPK-dependent phosphorylation pathways [86]. The differential expression of osmotic stress transcriptional factor (Ostf) was first revealed in isolated CCs and PVCs [80]. A study confirmed that cortisol directly acts on glycogen-rich cells in the gills of tilapia and regulates glycogen metabolism by promoting glycogen phosphorylase isoform (GPGG) mRNA expression [87]. To maintain their good health, the immune response can be generated by fish gills against external threats as the first line of immune defense [81]. The physiological and immune functions of tilapia gills are widely understood, but their functional heterogeneity at the single-cell scale has rarely been reported. One study performed single-cell RNA sequencing (scRNA-seq) on the gills of tilapia (*Oreochromis niloticus*) and identified 12 cell populations and analyzed their functional heterogeneity [88]. The single-cell datasets provide a reference for marker gene establishment of gills and serve as a platform for future studies investigating the physiological and immune function of gills.

The absorption rate of ten drugs was determined by a gill cell culture system [89]. The effect of ferric sulfate (FeSO_4_·7H_2_O) on the semi-permeable membrane culture of primary gill cells of rainbow trout was studied. This model was used to study the toxicity and drug resistance of gill epithelial cells [90]. In addition, the study has demonstrated that the cultured gill epithelia of rainbow trout can be optimized to exhibit tolerance towards seawater, enabling their utilization in toxicological assessments of pollutants suspended in seawater, thus simulating conditions found in marine ecosystems [81]. This optimized gill cell system represents a viable in vitro approach for conducting toxicological studies on marine ecosystems, thereby facilitating effective pollution control and management. Significant progress has been made in the assessment of fish toxicity of microalgal species using highly sensitive and reproducible fish RTgill-W1 gill cell lines [91]. The potential of the primary gill cell culture system was further demonstrated for environmental monitoring and biotransformation of organic compounds in branchial cells [92]. Previous studies have reported how to reconstruct and culture freshwater rainbow trout gill epithelium on a flat permeable membrane. The system can be used for freshwater gill physiology and toxicity studies, bioaccumulation studies, and environmental water quality monitoring [93].

### 2.9. Fish Intestinal Epithelial Cells

The intestine plays a pivotal role in fish physiology, serving as the primary site for digestion, nutrient absorption, and various crucial functions including osmoregulation, acid–base balance, and the excretion of specific metabolic byproducts [94]. Within the intestinal epithelium, monolayer cells possess dual functions, as they both absorb essential substances and serve as a defense against harmful ones. The luminal cells in the intestine are closely linked to the epithelial cells by a brush border, forming a relatively impermeable membrane [95]. These intestinal epithelial cells also act as vital barriers to protect the gut and can be influenced by dietary nutrition and environmental factors [96]. For example, in aquaculture, prebiotics are intricate carbohydrate molecules that are not directly digested by fish but are metabolized by the microbial community within the host gut. This process promotes the growth of “beneficial” bacterial species, subsequently enhancing fish performance. 

It is worth noting that gut cells may directly respond to these dietary components, contributing to research on fish gut health [97]. Additionally, cultured fish intestinal epithelial cells can release warning substances against predatory behavior [98]. In cases of deteriorating water quality and feed quality, many carnivorous fish become susceptible to bacterial intestinal diseases, posing challenges to the sustainable development of the aquaculture industry. A novel ex vivo culture method using primary intestinal epithelial cells from rainbow trout, Oncorhynchus mykiss, has been established and maintained over the long term. This in vitro system allows for the study of fundamental processes related to regional differences in dietary uptake and contaminant metabolism. It also provides a means to reduce the number of fish required for biological studies [99]. Some studies have employed transcriptome sequencing technology to comprehensively and rapidly obtain all transcript information during the exposure of intestinal epithelial cells of cyprinid carp to lipopolysaccharides (the main virulence factor of Gram-negative bacteria). This systematic analysis of gene expression changes provides a theoretical basis for understanding the molecular mechanisms underlying the regulation of intestinal epithelial cells in *Cyprinus carpio* by lipopolysaccharides [100].

### 2.10. Other Cells

For neuron cells, using electrophysiological analysis to isolate and culture neurons from the spinal cord of adult zebrafish, this new primary cell culture system provides in vitro methodology and new tools for neurophysiological research [101]. 

The fin cell line is very versatile because it has a high reproductive capacity and the success rate of cell migration is high [10,102]. At the same time, many fish cells are derived from fin tissue, such as primary cell cultures derived from the fin of the endangered Yangtze sturgeon (*Acipenser dabryanus*) [103]. Fish polyploid cells can be inferred by the karyotype of their fin cells. Three cell lines were established in fin primary culture of diploid crucian, triploid hybridization, and allotetraploids. And one marker gene, collagen type I alpha 1 chain (COL1A1), was used to further identify cell karyotype [104]. Primary goldfish culture cells were derived from the caudal fin and incubated with a temperature range of 20 to 35 ℃. The results indicated that goldfish cells were produced with different temperatures and tended to show stable physiological states related to temperature after temperature changes [105]. The establishment of a pomfret fin cell line would offer a beneficial in vitro tool for mechanism studies about pathogen interaction [106]. The koi fin cell line was established to develop effective diagnostic methods for detecting and monitoring viral infections in the case of outbreaks of koi herpes virus disease in India [107]. 

## 3. Common Methods for Fish Cell Isolation 

The development of fish cell culture is a relatively recent advancement compared to mammalian cell culture, and its methodologies have been adapted from those used in mammals. Nevertheless, there are notable differences. Temperature control in cell culture is a critical factor, closely tied to the temperature preferences of the fish species. Since fish are poikilotherms, their cell culture temperature range is broader than that of mammals. Passage frequency, occurring every 7 to 14 days in fish cells, is comparatively lower than the 4-day passage frequency of mammalian cells [108]. Fish cells also exhibit a slower metabolic rate, allowing them to be maintained in cell culture for short periods (several months) due to their sluggish metabolism in a relatively cool environment. For long-term preservation, storage in liquid nitrogen (−196 °C) is necessary. The isolation of primary fish cells primarily relies on methods developed for mammalian cells, including the tissue block method, mechanical crushing method, and enzyme digestion method. 

### 3.1. Enzymatic Digestion

The enzyme digestion method involves removing interstitial components, such as matrix and fibers, which can hinder cell growth. This results in the dispersion of cells into a suspension, facilitating better nutrient absorption and efficient elimination of metabolic waste. Using the enzyme digestion method eliminates issues related to selective cell growth due to mobility, allowing for the rapid acquisition of a significant number of representative cells within a short timeframe. Protein-digesting enzymes commonly used for in vitro cell isolation include trypsin, collagenase, lysozyme, pepsin, and neutral protease, with trypsin and collagenase being frequently employed. In the process of fish cell isolation and culture using enzymatic digestion, the fish are first sterilized by immersing them in 75% ethanol after anesthesia. The dissected tissues are then placed into a cell buffer, cut into small pieces in a Petri dish using a sharp scalpel, and passed through different micropipette tips. Afterward, the softened tissue is agitated and passed through a disposable mesh (100 µm). The remaining tissue on the mesh is subjected to collagenase treatment and sieving. The suspension containing the isolated cells undergoes centrifugation and washing before being placed into the culture medium [41,109]. For example, trypsin was used to isolate large yellow croaker (*Larimichthys crocea*) gonadal cells, establish cell lines, and explore gene expression [22].

### 3.2. Tissue Block Adhesion Method

This method is suitable for tissues with low volume or dense growth, such as fin strips, skin, muscle, and gonads, where mechanical or enzymatic hydrolysis may cause cellular damage. However, a limitation of the tissue block adhesion method lies in its selective adhesion to certain tissues and their subsequent outward migration and growth. Nonetheless, this method remains one of the commonly used, straightforward, and feasible primary cell culture techniques. It involves the aseptic separation of the target tissue, followed by the removal of impurities like membranes, lipids, and blood using a buffer solution. Sterile anatomical tools are used to divide the tissue block into 1 mm^3^ pieces, which are then inoculated at the bottom of a cell bottle. Subsequently, culture medium is added, and the bottle is transferred to an incubator for cultivation. It is crucial to master the adhesion time to prevent nutritional-deficiency-induced death of tissue cells [110]. Cells cultured using the tissue block adhesion method have the advantage of maintaining complete cell morphology and strong proliferation ability. For instance, Li et al. isolated and cultured grass goldfish (*Carassius auratus*) muscle primary cells using the tissue block adhesion method and established a new continuous fish cell line (CAM) [33].

### 3.3. Mechanical Crushing

The mechanical crushing method is suitable for tissues that have soft electrolytes, fewer fibrous components, and can withstand mechanical separation. Examples of such tissues include embryos, spleens, livers, adult fish brains, and soft tumors. This method involves breaking down and isolating tissue blocks into individual cells through various physical techniques, such as repeated grinding and filtration or tissue disruption using nylon mesh and stainless steel mesh. Subsequently, centrifugation is employed to obtain a single-cell suspension. While this approach allows for the rapid acquisition of a large number of cells, it is susceptible to causing mechanical damage. The tissue adhesion method, when used for primary cell culture, results in slow cellular migration and is applicable to only a limited range of tissues. On the other hand, the enzymatic digestion method carries a risk of cell damage during processing, prompting many researchers to prefer mechanical crushing as a viable alternative. This method was successfully utilized for the isolation of intestinal epithelial cells (IECs) from grass carp, yielding favorable results [111].

## 4. Perspectives and Future Developments

The transfection efficiency of gonadal cell lines from sole was up to 40%. Therefore, an effective method for transfection is imperative. In many vitro studies of primary cultured fish cells, the cultured living cells have characteristics that were similar to those of intact cells in vivo. However, important functional characteristics of cells may be lost in cell dispersion or the following culture, although primary cells can remain active for a long time [29]. Long-term cultured cells with a high proliferation rate can be easily frozen and thawed when necessary, which will offer a good basis for use. However, the primary cells can benefit from maintaining interactions and functional properties between primary cells. A major task in secondary cell culture is that they have the same properties as primary cells [18]. In vitro toxicity assessment and screening were widely used in cell lines due to their simple availability and cost effectiveness. Primary cell cultures have been criticized for the fact that they may experience significant genotype and phenotypic mutations, and the karyotype changes have an impact on gene expression and function [23] (Table 1).

**Table 1 biology-12-01454-t001:** Examples of primary cell culture listed in this paper.

Reference	Species	Tissue	Main Experimental Use	Culture Media	Serum	Antibiotics	Growth Factor, Hormone, and Others in Media
[11]	Marine medaka (*Oryzias melastigma*)	Pituitary, testis, and ovary	Rapid screening of environmental chemicals	L-15 medium	Fetal bovine serum	Fungizone, penicillin, and streptomycin	Insulin–transferrin–selenium-A, GlutaMAX
[29]	Atlantic cod (*Gadus morhua*)	Pituitary	Optimizing the conditions of cell culture	M199 medium	Newborn calf serum	None	None
[27]	Tilapia (*Oreochromis niloticus*)	Pituitary	Signaling pathway validation	M199 medium	Fetal bovine serum	None	None
[23]	Medaka (*Oryzias melastigma*)	Pituitary	Rapid screening of environmental silver nanoparticles	M199 medium	Fetal bovine serum	Fungizone	1 × GlutaMAX
[25]	Atlantic cod (*Gadus morhua*)	Pituitary	Gene function analysis	L-15 medium	Newborn calf serum	None	None
[26]	Orange-spotted grouper (*Epinephelus coioides*)	Pituitary	Signaling pathway validation	L-15 medium	Bovine serum albumin	Penicillin, streptomycin	Sodium chloride
[10]	Medaka (*Oryzias latipes*)	Testis	Research of hormone metabolism	L-15 medium	Fetal calf serum	Penicillin, streptomycin	Steroid-free Ultroser SF
[87]	Tilapia (*Oreochromis niloticus*)	Testis	Analysis of spermatogenesis	L-15 medium	Fetal bovine serum	Penicillin, streptomycin	None
[103]	Creekchub (*Semotilus atromaculatus*)	Skin	Rapid screening of environmental chemicals	L-15 medium	Fetal bovine serum	Penicillin, Fungizone, kanamycin, and tetracycline	None
[31]	Gilthead sea bream (*Sparus aurata*)	Muscle	Signaling pathway validation	DMEM	Horse serum	Penicillin, streptomycin, fungizone, gentamycin	None
[34]	Brown trout (*Salmo trutta*)	Muscle	Signaling pathway validation	DMEM	Horse serum	Penicillin, streptomycin, amphotericin, and gentamycin	Poly-L-lysine, laminin
[30]	Rainbow trout (*Oncorhynchus mykiss*)	Muscle	Gene function analysis	DMEM	Fetal bovine serum	Penicillin, streptomycin	100 nM of trout IGF1
[32]	Rainbow trout (*Oncorhynchus mykiss*)	Muscle	Characterization of proliferation and differentiation	DMEM	Horse serum	Penicillin, streptomycin, Fungizone	None
[3]	Orange-spotted grouper (*Epinephelus coioides*)	Liver	Evaluation of the nonylphenol-induced oxidative stress	L-15 medium	Fetal bovine serum	Penicillin, streptomycin	None
[41]	Rainbow trout (Oncorhynchus mykiss)	Liver	Toxicology research	L-15 medium	None	Amphotericin, streptomycin, penicillin	EE2, L-glutamine
[40]	Arctic char (*Salvelinus alpinus*)	Liver	Screening environmental contaminants	L-15 medium	None	Penicillin, streptomycin, amphotericin	L-glutamine
[42]	Brown trout (*Salmo trutta*)	Liver	Signaling pathway validation	L-15 medium	Fetal bovine serum	Streptomycin, penicillin	Poly-L-lysine (300 μg/mL)
[4]	Orange-spotted grouper (*Epinephelus coioides*)	Liver	Toxicology research	L-15 medium	Fetal bovine serum	Penicillin, streptomycin	None
[47]	Yellow catfish (*Pelteobagrus fulvidraco*)	Liver	Research of hormone metabolism	M199 medium	Fetal bovine serum	Penicillin, streptomycin	L-glutamine
[49]	Brown trout (Salmo trutta)	Liver	Gene function analysis	L-15 medium	Fetal bovine serum	Streptomycin, penicillin	None
[51]	White sturgeon (*Acipenser transmontanus*)	Liver	Test intracellular pH compensation	α-minimum essential medium	Fetal bovine serum	Penicillin, streptomycin, Fungizone	None
[43]	Rainbow trout (*Oncorhynchus mykiss*)	Liver	Comparative cytotoxicity study of silver nanoparticles	L-15 medium	Fetal bovine serum	Penicillin, streptomycin	L-glutamine, sodium pyruvate, NEAA
[44]	Orange-spotted grouper (*Epinephelus coioides*)	Liver	Toxicology research	DMEM/F12 medium	Fetal bovine serum	Penicillin, streptomycin	None
[101]	Grass carp (*Ctenopharyngodon* *idellus*)	Intestine	Signaling pathway validation	DMEM	Fetal bovine serum	Penicillin, gentamycin, amphotericin B, gentamycin sulfate	None
[104]	Rainbow trout (*Oncorhynchus mykiss*)	Intestine	Cell culture method	Hanks’ balanced salt solution	Bovine serum albumin	None	None
[65]	Goldfish (*Carassius auratus* L.)	Kidney	Gene function analysis	MGFL-15 medium	Heat-inactivated carp serum	Penicillin, streptomycin	Mechano Growth Factor
[66]	Gilthead seabream (*Sparus aurata* L.)	Kidney	Immunological research	RPMI 1640 medium	Fetal bovine serum	Penicillin, streptomycin	Heparin
[72]	Goldfish (*Carassius auratus* L.)	Kidney	Immunological research	NMGFL-15 medium	Calf serum and carp serum	None	Interleukin-3, granulocyte–macrophage-colony-stimulating factor
[60]	Rainbow trout (*Oncorhynchus mykiss*) and gilthead sea bream (*Sparus aurata*)	Kidney	Immunological research	DMEM	None	None	Antagonist receptors and/or hormones
[6]	Wolf fish *(Hoplias malabaricus)*	Kidney	Immunological research	L-15 medium	Fetal bovine serum	Penicillin, streptomycin	None
[5]	Red carp (*Cyprinus carpio*)	Kidney	Immunological research	Hank’s balanced salt solution	None	Penicillin, streptomycin	Heparin
[64]	European eel (*Anguilla anguilla*)	Kidney	Immunological research	L-15 medium	Bovine serum albumin	None	Poly-L-lysine
[68]	Rainbow trout (*Oncorhynchus mykiss*)	Adipocyte	Toxicology research	L-15 medium	Fetal bovine serum	Antibiotic/antimycotic solution	Insulin
[69]	Rainbow trout (*Oncorhynchus mykiss*)	Adipocyte	Research of hormone metabolism	Krebs–HEPES buffer	Fetal bovine serum	Collagenase type II	Insulin
[67]	Rainbow trout (*Oncorhynchus mykiss*)	Adipocyte	Gene function analysis	Krebs–HEPES buffer	Bovine serum albumin	1% antibiotic/antimycotic solution	L-glutamine, insulin
[106]	Zebrafish (*Danio rerio*)	Brain	Electro-physiological studies	L-15 medium	Fetal bovine serum	Penicillin, streptomycin	Ofloxacin
[74]	Giant groupers (*Epinephelus lanceolatus*)	Brain	Immunological research	L-15 medium	Fetal bovine serum	Penicillin, streptomycin	None
[48]	Half-smooth tongue sole (*Cynoglossus semilaevis*)	Brain	Gene function analysis	L-15 medium	Bovine serum albumin	Penicillin, streptomycin	Palmitic acid
[75]	*Sole* (*Solea senegalensis*)	Brain	Immunological research	L-15 medium	Fetal bovine serum	Gentamicin	Glutamine
[73]	Sea bream (*Sparus aurata*)	Bone	Research of hormone metabolism	DMEM	Fetal bovine serum	Antibiotic/antimycotic solution	L-ascorbic acid, β-glycerophosphate
[71]	Gilthead sea bream (*Sparus aurata*)	Bone	Gene function analysis	DMEM	Fetal bovine serum	Antibiotic/antimycotic solution	NaCl, porcine insulin, dexamethasone
[78]	Common carp (*Cyprinus carpio*)	Fin	Immunological research	DMEM	Fetal bovine serum	Phosphate, streptomycin	FGF
[97]	Rainbow trout (*Oncorhynchus mykiss*)	Gill	Environmental monitoring of urban streams	L-15 medium	Fetal bovine serum	Penicillin, streptomycin, gentamicin	None
[81]	Rainbow trout (*Oncorhynchus mykiss*)	Gill	Salinity/water regulation	Hanks’ balanced salt solution	Fetal bovine serum	Penicillin, Streptomycin, and amphotericin B	Glutamine
[82]	Puffer fish (*Tetraodon nigroviridis*)	Gill	Gene function analysis	L-15 medium	Fetal bovine serum	Penicillin, streptomycin, gentamicin	L-glutamine, cortisol
[90]	Japanese eels (*Anguilla japonica)*	Gill	Salinity/water regulation	L-15 medium	Fetal bovine serum	Penicillin, streptomycin, and Fungizone	None
[95]	Rainbow trout (*Oncorhynchus mykiss*)	Gill	Toxicology research	L-15 medium	Fetal bovine serum	Penicillin, streptomycin, amphotericin B	Glycine

Advanced in vitro culture from tissues of different origins includes three-dimensional organoid microstructures that may mimic conditions in vivo. Direct measurements of oxygen gradients in a spheroid culture system were used in electron parametric resonance oximetry, which provides an elegant, widely applicable approach to directly characterize spheroid (and another organoid) cultures in biomedical and toxicological research [112]. In another application in retinal organoids, the cell–cell interactions are necessary to investigate the formation of retinal layers. The dissociated zebrafish retinal progenitors were cultured in agarose microwells, forming tight retinal organoids within these wells [113]. 

Cell lines, especially stem cells, have always been the first choice for cell-level research due to their advantages of easy culture, variety, and considerable yield. It is noteworthy that some cell lines have been involved in important milestones, such as the pioneering work on stem cells in the medaka model [114,115,116].

## 5. Conclusions

In conclusion, the review emphasizes the significance of fish primary cell culture as a powerful tool for profiling the physiological roles of fish cells. The utilization of fish primary cell culture techniques has previously facilitated the identification of emerging viruses and is now extensively employed in research on environmental toxicology, immunology, fish physiology, and germplasm conservation. This technique has revolutionized our understanding of cellular processes in fish species, enabling researchers to investigate cellular responses, unravel intricate molecular mechanisms, and explore novel avenues for fish health management, aquaculture optimization, and environmental risk assessment.

Therefore, it is crucial to standardize and diversify fish primary cell separation technology and cell culture conditions for the optimal application of fish cells. Cell culture technology has permeated every facet of life science and holds immense potential for further development.

## Figures and Tables

**Figure 1 biology-12-01454-f001:**
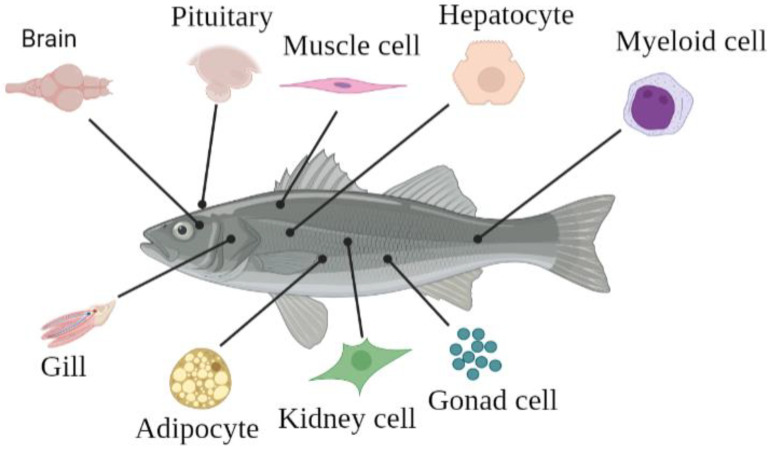
General view of primary cell culture from various tissues or organs including brain, pitui-tary, muscle, liver, bone, gonad, kidney, fat, and gill.

**Figure 2 biology-12-01454-f002:**
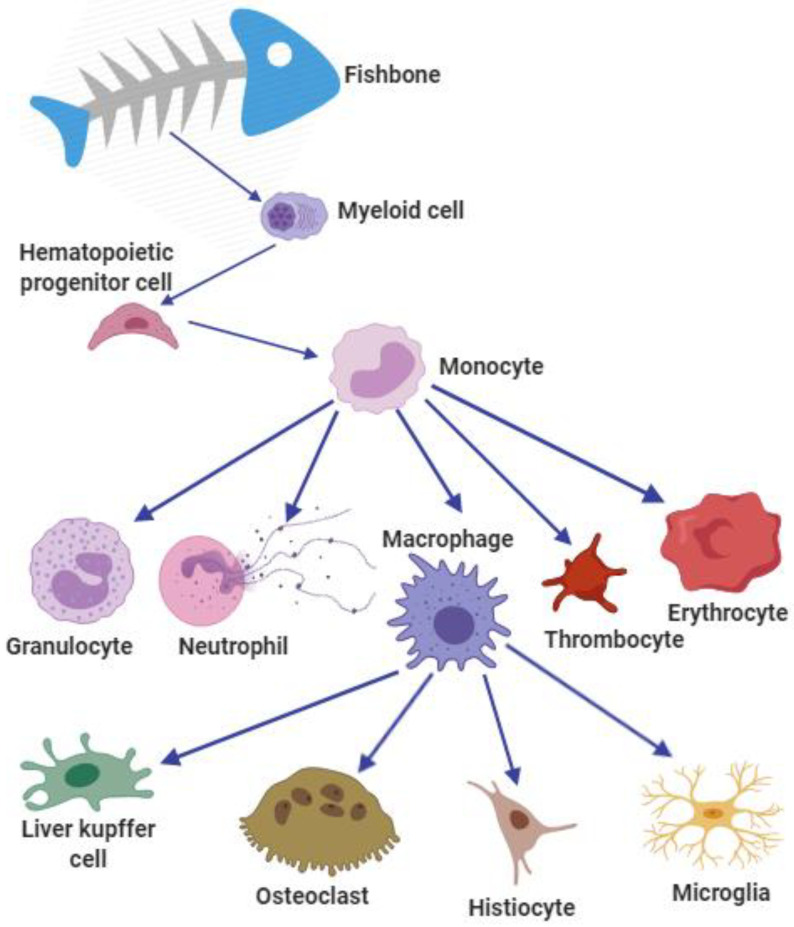
The cells derived from fishbone. Myeloid cells develop into hematopoietic progenitor cells following by monocytes. Then, monocytes evolve into granulocytes, neutrophils, macrophages, thrombocytes, and erythrocytes. Macrophages ultimately transform into liver Kupffer cells, osteoclasts, histiocytes, and microglia.

## Data Availability

No new data were created or analyzed in this study. Data sharing is not applicable to this article.

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
