# Peer review of "Profiling the Physiological Roles in Fish Primary Cell Culture"

_biology, 2023, doi:10.3390/biology12121454_

Round 1
Reviewer 1 Report
Comments and Suggestions for Authors
Dear authors,
After exhaustive revision I have some comments:
1.- introduction need more work, for example to clarify differences between primary cell culture and cell line.
2.- I like the presentation, but I think that need more work for expample Kidney, in fish has 2 portions Head kidney: it is more endocrine (cortisol) and immune than psoterior kidney, it is more osmotic and metabolic than anterior portion. I was looking for information and please to cheack these papers:
Nualart, et al. Immune Transcriptional Response in Head Kidney Primary Cell Cultures Isolated from the Three Most Important Species in Chilean Salmonids Aquaculture. Biology 2023, 12, 924. https://doi.org/10.3390/biology12070924
Pontigo, 2021, Growth hormone (GH) and growth hormone release factor (GRF) modulate the immune response in the SHK-1 cell line and leukocyte cultures of head kidney in Atlantic salmon. General and Comparative Endocrinology 300 (2021) 113631. https://doi.org/10.1016/j.ygcen.2020.113631
Gills, more information and examples are necessary.
Intestine? why has not an independent paragraph?
Comments on the Quality of English LanguageThe lenguage need minor check.
Author Response
Dear authors,
After exhaustive revision I have some comments:
1.- introduction need more work, for example to clarify differences between primary cell culture and cell line.
Response: Thank you so much for your valuable feedback. We have taken them into consideration and made the necessary revisions to the manuscript. We reviewed the reference and added the introduction, which mainly included the differences between primary cells and cell lines, their advantages and disadvantages, the necessity of primary cell culture and its significance to the development of aquaculture.
The differences between primary cell culture and cell line (line 73 - line 100)
Primary cells and cell lines both have advantages and disadvantages. The growth rate of primary cells is notably slow, and conventionally, the initial generation of primary cell culture and its subsequent passages up to the 10th generation collectively form the primary culture. Beyond approximately 10 generations, primary cultured cells present a formidable challenge for propagation. Cellular growth tends to stagnate, and most cells undergo senescence and apoptosis. Nevertheless, a small fraction of cells somehow endure this crisis, persisting in propagation. Some of these cells may even undergo genetic alterations and acquire malignant characteristics, allowing for indefinite passage ability, which is referred to as a cell line. Primary cells, isolated directly from animal tissues through enzymatic or mechanical methods, are often considered more biologically relevant than cell lines. This is because their biological responses may closely mimic in vivo conditions, bringing researchers closer to the "truth"[1]. Early cultures of primary cells can offer a more accurate model of in vivo tissues [2]. However, the production of short-term primary cultures faces challenges related to the reproducibility of initiation and the homogeneity of cultures, limiting their applications [3]. The advantages of easy culture, diversity, and substantial yield have positioned cell lines as the preferred choice for cellular-level research. Nevertheless, it is important to note that continuous culturing of cell lines can lead to mutations over time, potentially altering both their genotype and phenotype, which in turn may impact experimental outcomes. Primary cells do not encounter these problems. For instance, cells obtained from EK293 transfected with adenovirus exhibited characteristics closer to immature neurons[4]. The MDA-435 cell line, long utilized as a representative model for breast cancer, has recently raised questions about its true identity, with accumulating evidence suggesting its potential classification as a melanoma cell[5]. In summary, for experiments that cannot be conducted within living organisms, primary cells can maintain a high level of biological characteristics and thus offer a partial solution to this issue. Vitro cell culture is a fascinating experiment method as the cells can maintain their interaction, polarity and topology [6].
Reference:
[1]Ager-Wick E, Hodne K, Fontaine R, von Krogh K, Haug TM, Weltzien FA: Preparation of a High-quality Primary Cell Culture from Fish Pituitaries. Jove-Journal of Visualized Experiments 2018(138).
[2]Lakra WS, Swaminathan TR, Joy KP: Development, characterization, conservation and storage of fish cell lines: a review. Fish Physiol Biochem 2011, 37(1):1-20.
[3]Speirs V, Green AR, Walton DS, Kerin MJ, Fox JN, Carleton PJ, Desai SB, Atkin SL: Short-term primary culture of epithelial cells derived from human breast tumors. Journal of Cancer 1998, 78(11):1421-1429.
[4]Shaw G, Morse S, Ararat M, Graham FL: Preferential transformation of human neuronal cells by human adenoviruses and the origin of HEK 293 cells. Journal of Ficial Publication of the Federation of American Societies for Experimental Biology 2002, 16(8):869-871.
[5]Korch C, Hall EM, Dirks WG, Ewing M, Faries M, Varella-Garcia M, Robinson S, Storts D, Turner JA, Wang Y et al: Authentication of M14 melanoma cell line proves misidentification of MDA-MB-435 breast cancer cell line. International Journal of Cancer 2018, 142(3):561-572.
[6]Song M, Gutzeit HO: Effect of 17-alpha-ethynylestradiol on germ cell proliferation in organ and primary culture of medaka (Oryzias latipes) testis. Development Growth & Differentiation 2003, 45(4):327-337.
Its significance to the development of aquaculture. (line 105-line 109, line 127-line 138,)
The principles and techniques of fish cell culture draw heavily from mammalian cell culture practices. However, when compared to mammalian cell culture, fish cell culture presents some distinct characteristics. Notably, fish cell culture boasts several advantages including its ability to adapt to a wide range of temperatures, higher tolerance to hypoxia, and the ease of maintaining cell cultures for extended durations. The growing economic significance of aquaculture has driven increased research into the healthy development and pathological occurrence of fish, with particular attention being paid to the prevention and control of freshwater and seawater diseases, as well as environmental issues. The advancement of cell culture technology has played a pivotal role in enabling research in genomics, the study of virus-host interaction studies, bacteria identification, heavy metal toxicity analysis, and stem cell function research. To our knowledge, no review has brought the information on the primary cell cultures from different fish tissues. Thus, we report on the progress of primary cell cultures derived from the gonad, pituitary, muscle, liver, kidney, adipocyte, myeloid, brain, fin and gill and the historical background, practical applications, and cultivation techniques employed in fish cell culture in this review. It is expected to provide some valuable references for the development of fish primary cell culture and the establishment of cell lines.
2.- I like the presentation, but I think that need more work for example Kidney, in fish has 2 portions Head kidney: it is more endocrine (cortisol) and immune than psoterior kidney, it is more osmotic and metabolic than anterior portion. I was looking for information and please to cheack these papers:
Nualart, et al. Immune Transcriptional Response in Head Kidney Primary Cell Cultures Isolated from the Three Most Important Species in Chilean Salmonids Aquaculture. Biology 2023, 12, 924. https://doi.org/10.3390/biology12070924
Pontigo, 2021, Growth hormone (GH) and growth hormone release factor (GRF) modulate the immune response in the SHK-1 cell line and leukocyte cultures of head kidney in Atlantic salmon. General and Comparative Endocrinology 300 (2021) 113631. https://doi.org/10.1016/j.ygcen.2020.113631
Response: Thank you for your suggestion. We have carefully reviewed the suggested studies and find them highly relevant. Insights from these studies have been integrated into our research with proper references kindly refer to the revised manuscript for details. (line 281-line 309)
On the other hand, the head kidney primarily plays a crucial role in the production of macrophages in fish. These macrophages, derived from the head kidney, exhibit the capacity to phagocytize, produce radicals, and polarize into either innately activated or alternatively activated macrophages [1]. Consequently, they play a pivotal role in both the innate and acquired immune responses of fish[2]. In an exciting development, a continuous cell line has been established, characterized, and isolated from the head kidney of a large yellow croaker. This cell line holds significant potential as a valuable tool for studying immune-related genes and functions [3]. Furthermore, a study involving seabass head kidney in vitro research has shed light on the immunomodulatory effects of various amino acids. This underscores the potential for developing an immune-nutrition strategy[4]. The aquaculture industry has suffered significant economic losses due to bacterial and viral infections. In order to mitigate these losses, cell cultures offer a viable alternative to in vivo experimentation. A study was conducted to kidney primary cell cultures from three Chilean salmonids, namely Salmo salar, Oncorhynchus kisutch, and Oncorhynchus mykiss, the aim of the study was to characterize the response to bacterial and viral stimuli by evaluating various markers associated with both innate and adaptive immune responses[5]. The morphology of primary cultured cells from the head kidney of rainbow trout (Oncorhynchus mykiss) is depicted in Fig.2E. In a study, the time course stimulation with GH and GRF was described in two experimental models: the first model involved the SHK-1 cell line derived from primary cultures of adherent cells from Atlantic salmon (Salmo salar) head kidney, which exhibited phagocytic characteristics; and the second model utilized leukocytes isolated from the head kidney of Atlantic salmon. The results indicated differential regulation between these two models, providing a better understanding of the independent action of GH on the immune system[6].
Reference
[1] Joerink M, Ribeiro CMS, Stet RJM, Hermsen T, Savelkoul HFJ, Wiegertjes GF: Head kidney-derived macrophages of common carp (Cyprinus carpio L.) show plasticity and functional polarization upon differential stimulation. Journal of Immunology (Baltimore, Md : 1950) 2006, 177(1):61-69.
[2] Xu Z, Takizawa F, Parra D, Gomez D, Jorgensen LvG, LaPatra SE, Sunyer JO: Mucosal immunoglobulins at respiratory surfaces mark an ancient association that predates the emergence of tetrapods. Nture Comunications. 2016, 7.
[3] Wang XH, Wang KR, Nie P, Chen XH, Ao JQ: Establishment and characterization of a head kidney cell line from large yellow croaker Pseudosciaena crocea. Journal of Fish Biology 2014, 84(5):1551-1561.
[4] Azeredo R, Serra CR, Oliva-Teles A, Costas B: Amino acids as modulators of the European seabass, Dicentrarchus labrax, innate immune response: an in vitro approach. Science Reports 2017, 7.
[5] Nualart DPP, Dann F, Oyarzun-Salazar R, Morera FJJ, Vargas-Chacoff L: Immune Transcriptional Response in Head Kidney Primary Cell Cultures Isolated from the Three Most Important Species in Chilean Salmonids Aquaculture. Biology-Basel 2023, 12(7).
[6] Pontigo JP, Vargas-Chacoff L: Growth hormone (GH) and growth hormone release factor (GRF) modulate the immune response in the SHK-1 cell line and leukocyte cultures of head kidney in Atlantic salmon. General and Comparative Endocrinology 2021, 300.
Gills, more information and examples are necessary.
Response: Thank you for bringing this matter to our attention. We acknowledge the insufficiency of descriptions pertaining to fish gill cells. In response to your reminder, we have included additional examples of fish gill primary cell culture in relation to immunity and toxicology. (line 408-line 413, line 422-line 432, line 436-line 441)
A primary culture of rainbow trout (Oncorhynchus mykiss) gill epithelial cells in Fig. 2G[1]. The primary culture of fish gill cells can provide a functional, cell diversified and model culture platform in vitro. In a study to prolong the viability of primary gill cell cultures in rainbow trout, a method was established to prolong the viability of cultures during prolonged exposure to water, with the utility of this model extending to a variety of longer-term exposure scenarios[2].
A study confirmed that cortisol directly acts on glycogen-rich cells in the gills of tilapia and regulates glycogen metabolism by promoting glycogen phosphorylase isoform (GPGG) mRNA expression[3]. To maintain their good health, immune response can be generated by fish gills against external threats as the first line of immune defense[4]. The physiological and immune functions of tilapia gills have been widely understood, but their functional heterogeneity at the single-cell scale has rarely been reported. One study performed single-cell RNA sequencing (scRNA-seq) on the gills of tilapia (Oreochromis niloticus) and identified 12 cell populations and analyzed their functional heterogeneity[5]. The single cell datasets provide a reference for marker gene establishment of gills and serve as a platform for future studies investigating the physiological and immune function of gills.
In addition, the study has demonstrated that the cultured gill epithelia of rainbow trout can be optimized to exhibit tolerance towards seawater, enabling their utilization in toxicological assessments of pollutants suspended in seawater, thus simulating conditions found in marine ecosystems[6]. This optimized gill cell system represents a viable in vitro approach for conducting toxicological studies on marine ecosystems, thereby facilitating effective pollution control and management.
Reference
[1]Witters, H.; Berckmans, P.; Vangenechten, C. Immunolocalization of Na+, K+-ATPase in the gill epithelium of rainbow trout, Oncorhynchus mykiss. Cell Tissue Res 1996, 283, 461-468,
[2]Maunder RJ, Baron MG, Owen SF, Jha AN: Investigations to extend viability of a rainbow trout primary gill cell culture. Ecotoxicology 2017, 26(10):1314-1326.
[3] Wu CY, Lee TH, Tseng DY: Glucocorticoid Receptor Mediates Cortisol Regulation of Glycogen Metabolism in Gills of the Euryhaline Tilapia (Oreochromis mossambicus). Fishes 2023, 8(5).
[4] Gomez D, Sunyer JO, Salinas I: The mucosal immune system of fish: The evolution of tolerating commensals while fighting pathogens. Fish & Shellfish Immunology 2013, 35(6):1729-1739.
[5] Zheng S, Wang W-X: Physiological and immune profiling of tilapia Oreochromis niloticus gills by high-throughput single-cell transcriptome sequencing. Fish & Shellfish Immunology 2023, 141:109070-109070.
[6] Bawa-Allah KA, Otitoloju A, Hogstrand C: Cultured rainbow trout gill epithelium as an in vitro method for marine ecosystem toxicological studies. Heliyon 2021, 7(9).
Intestine? why has not an independent paragraph?
Response: We are grateful for your thorough review of our manuscript. We acknowledge the necessity of listing intestinal cells in a separate paragraph and have accordingly incorporated this part into the original draft as an independent section, thereby enhancing its logical structure. We sincerely appreciate your valuable input. (line 450-line 478)
Fish intestinal epithelial cells
The intestine plays a pivotal role in fish physiology, serving as the primary site for digestion, nutrient absorption, and various crucial functions including osmoregulation, acid-base balance, and the excretion of specific metabolic byproducts [1]. Within the intestinal epithelium, monolayer cells possess dual functions, as they both absorb essential substances and serve as a defense against harmful ones. The luminal cells in the intestine are closely linked to the epithelial cells by a brush border, forming a relatively impermeable membrane [2]. These intestinal epithelial cells also act as vital barriers to protect the gut and can be influenced by dietary nutrition and environmental factors [3]. For example, in aquaculture, prebiotics are intricate carbohydrate molecules that aren't directly digested by fish but are metabolized by the microbial community within the host gut. This process promotes the growth of "beneficial" bacterial species, subsequently enhancing fish performance.
It's worth noting that gut cells may directly respond to these dietary components, contributing to research on fish gut health [4]. Additionally, cultured fish intestinal epithelial cells can release warning substances against predatory behavior [5]. In cases of deteriorating water quality and feed quality, many carnivorous fish become susceptible to bacterial intestinal diseases, posing challenges to the sustainable development of the aquaculture industry. A novel ex vivo culture method using primary intestinal epithelial cells from rainbow trout, Oncorhynchus mykiss, has been established and maintained over the long term. This in vitro system allows for the study of fundamental processes related to regional differences in dietary uptake and contaminant metabolism. It also provides a means to reduce the number of fish required for biological studies [6]. Some studies have employed transcriptome sequencing technology to comprehensively and rapidly obtain all transcript information during the exposure of lipopolysaccharides (the main virulence factor of Gram-negative bacteria) to intestinal epithelial cells of Cyprinid carp. This systematic analysis of gene expression changes provides a theoretical basis for understanding the molecular mechanisms underlying the regulation of intestinal epithelial cells in Cyprinus carpio by lipopolysaccharides [7].
Reference
[1] Bieczynski F, Painefilu JC, Venturino A, Luquet CM: Expression and Function of ABC Proteins in Fish Intestine. Frontiers in Physiology 2021, 12.
[2] Guenzel D, Yu ASL: Claudins and the modulation of tight junction permeability. Physiological Reviews 2013, 93(2):525-569.
[3] Chen J, Zhang D, Tan Q, Liu M, Hu P: Arginine affects growth and integrity of grass carp enterocytes by regulating TOR signaling pathway and tight junction proteins. Fish Physiology and Biochemistry 2019, 45(2):539-549.
[4] Porter D, Peggs D, McGurk C, Martin SAM: Immune responses to prebiotics in farmed salmonid fish: How transcriptomic approaches help interpret responses. Fish & Shellfish Immunology 2022, 127:35-47.
[5] Hintz HA, Weihing C, Bayer R, Lonzarich D, Bryant W: Cultured fish epithelial cells are a source of alarm substance. MethodsX 2017, 4:480-485.
[6] Langan LM, Owen SF, Jha AN: Establishment and long-term maintenance of primary intestinal epithelial cells cultured from the rainbow trout, Oncorhynchus mykiss. Biology Open 2018, 7(3).
[7] Low C, Wadsworth S, Burrells C, Secombes CJ: Expression of immune genes in turbot (Scophthalmus maximus) fed a nucleotide-supplemented diet. Aquaculture 2003, 221(1-4):23-40.

Reviewer 2 Report
Comments and Suggestions for Authors
Overall, the manuscript is quite written introducing different types of cells from fish. However, I would suggest including some writing of fish cell isolation, for example, how fish cell isolation is different compared to mamallian cell isolation.
The second thing I would like to point out here is that for many abbreviations in the manuscript, the authors did not include the full name. For example, RNAi should be written as" RNAi (RNA interference)",
The third suggestion is that for the table of media, I would suggest to add a column indicating the growth factor or hormone they may add to media.
Comments on the Quality of English LanguageThe English is OK. However, some minor parts may need to be improved as they are not academic enough.
Author Response
Overall, the manuscript is quite written introducing different types of cells from fish. However, I would suggest including some writing of fish cell isolation, for example, how fish cell isolation is different compared to mammalian cell isolation.
Response: Thank you for your kind reminder. We acknowledge the significance of primary cell separation in our manuscript. Following your suggestion, we have incorporated a dedicated section that outlines the fundamental techniques and references for fish primary cell isolation, as well as highlighting the similarities and differences with mammalian cell isolation and culture. This addition has significantly enhanced the quality of our manuscript. Once again, we sincerely appreciate your valuable advice. (line 498-line 559)
Common methods for cell isolation in fish
The development of fish cell culture is a relatively recent advancement compared to mammalian cell culture, and its methodologies have been adapted from those used in mammals. Nevertheless, there are notable differences. Temperature control in cell culture is a critical factor, closely tied to the temperature preferences of the fish species. Since fish are poikilotherms, their cell culture temperature range is broader than that of mammals. Passage frequency, occurring every 7 to 14 days in fish cells, is comparatively lower than the 4-day passage frequency of mammalian cells [1]. Fish cells also exhibit a slower metabolic rate, allowing them to be maintained in cell culture for short periods (several months) due to their sluggish metabolism in a relatively cool environment. For long-term preservation, storage in liquid nitrogen (-196°C) is necessary. The isolation of primary fish cells primarily relies on methods developed for mammalian cells, including the tissue block method, mechanical crushing method, and enzyme digestion method.
Enzymatic digestion
The enzyme digestion method involves removing interstitial components, such as matrix and fibers, which can hinder cell growth. This results in the dispersion of cells into a suspension, facilitating better nutrient absorption and efficient elimination of metabolic waste. Using the enzyme digestion method eliminates issues related to selec-tive cell growth due to mobility, allowing for the rapid acquisition of a significant number of representative cells within a short timeframe. Protein-digesting enzymes commonly used for in vitro cell isolation include trypsin, collagenase, lysozyme, pep-sin, and neutral protease, with trypsin and collagenase being frequently employed. In the process of fish cell isolation and culture using enzymatic digestion, the fish are first sterilized by immersing them in 75% ethanol after anesthesia. The dissected tissues are then placed into a cell buffer, cut into small pieces in a Petri dish using a sharp scalpel, and passed through different micropipette tips. Afterward, the softened tissue is agitat-ed and passed through a disposable mesh (100 µm). The remaining tissue on the mesh is subjected to collagenase treatment and sieving. The suspension containing the isolat-ed cells undergoes centrifugation and washing before being placed into the culture medium [2, 3]. For example, trypsin was used to isolate large yellow croaker (Larimichthys crocea) gonadal cells, establish cell lines and explore gene expression[4].
Tissue block adhesion method
This method is suitable for tissues with low volume or dense growth, such as fin strips, skin, muscle, and gonads, where mechanical or enzymatic hydrolysis may cause cellular damage. However, a limitation of the tissue block adhesion method lies in its selective adhesion to certain tissues and their subsequent outward migration and growth. Nonetheless, this method remains one of the commonly used, straightforward, and feasible primary cell culture techniques. It involves the aseptic separation of the target tissue, followed by the removal of impurities like membranes, lipids, and blood using a buffer solution. Sterile anatomical tools are used to divide the tissue block into 1 mm3 pieces, which are then inoculated at the bottom of a cell bottle. Subsequently, culture medium is added, and the bottle is transferred to an incubator for cultivation. It's crucial to master the adhesion time to prevent nutritional deficiency-induced death of tissue cells [5]. Cells cultured using the tissue block adhesion method have the advantage of maintaining complete cell morphology and strong proliferation ability. For instance, Li et al. isolated and cultured grass goldfish (Carassius auratus) muscle primary cells using the tissue block adhesion method and established a new continuous fish cell line (CAM)(Carassius auratus) [6].
Mechanical crushing
The mechanical crushing method is suitable for tissues that have soft electrolytes, fewer fibrous components, and can withstand mechanical separation. Examples of such tissues include embryos, spleens, livers, adult fish brains, and soft tumors. This method involves breaking down and isolating tissue blocks into individual cells through various physical techniques, such as repeated grinding and filtration or tissue disruption using nylon mesh and stainless steel mesh. Subsequently, centrifugation is employed to obtain a single-cell suspension. While this approach allows for the rapid acquisition of a large number of cells, it is susceptible to causing mechanical damage. The tissue adhesion method, when used for primary cell culture, results in slow cellular migration and is applicable to only a limited range of tissues. On the other hand, the enzymatic digestion method carries a risk of cell damage during processing, prompting many researchers to prefer mechanical crushing as a viable alternative. This method was successfully utilized for the isolation of intestinal epithelial cells (IECs) from grass carp, yielding favorable results [7].
Reference
[1] Wolf K: Some Recent Developments and Applications of Fish Cell and Tissue Culture. The Progressive Fish-Culturist 1965, 27(2):67-74
[2] Kiraççakali AN, Oğuz AR: Determination of cytotoxic, genotoxic, and oxidative damage from deltamethrin on primary hepatocyte culture of Lake Van fish, Alburnus tarichi. Chemistry and Ecology 2020, 36(7):651-662.
[3] Kelly SP, Fletcher M, Pärt P, Wood CM: Procedures for the preparation and culture of 'reconstructed' rainbow trout branchial epithelia. Methods Cell Sci 2000, 22(2-3):153-163.
[4] Xu Y, Zhong Z, Zhang Z, Feng Y, Zhao L, Jiang Y, Wang Y: Establishment and characterization of the gonadal cell lines derived from large yellow croaker (Larimichthys crocea) for gene expression studies. Aquaculture 2022, 546.
[5] Deng Q, Liu L, Tang R, Xian D, Zhong J: A newly improved method of primary cell culture: Tissue block with continuous adhesion subculture in skin fibroblast. Acta Histochem 2023, 125(7):152090.
[6] Li N, Guo L, Guo H: Establishment, characterization, and transfection potential of a new continuous fish cell line (CAM) derived from the muscle tissue of grass goldfish (Carassius auratus). In Vitro Cell Dev Biol Anim 2021, 57(9):912-931.
[7] Yao SB YY, Cai CF, Yao LJ, Xu F, Liu M, Xiao PZ: Isolation and primary culture of intestinal mucosal epithelial cells of grass carp. Journal of Shanghai Ocean University 2013, 22:33-41.
The second thing I would like to point out here is that for many abbreviations in the manuscript, the authors did not include the full name. For example, RNAi should be written as" RNAi (RNA interference)",
Response: Thanks for your good advice. We have changed “RNAi " to “RNAi (RNA interference)” in the manuscript and other similar errors have been modified (line 271). For example, Cu-NPs (Copper nanoparticles) in line 251, IGF (insulin-like growth factor) in line 366, and so on.
The third suggestion is that for the table of media, I would suggest to add a column indicating the growth factor or hormone they may add to media.
Response: Thank you for your valuable suggestion. We have incorporated it by including an additional column in the table by referring to the original reference, as per your recommendation, which explains the growth factors, hormones and others added to the culture medium, so as to make the table more comprehensive. (line 574)

Round 2
Reviewer 1 Report
Comments and Suggestions for Authors
Dear Authors
I'm really happy to read a new version, it was improved a lot and for me it is Ok.
Author Response
Many thanks for your good comments.